# Accuracy of Segmented Le Fort I Osteotomy with Virtual Planning in Orthognathic Surgery Using Patient-Specific Implants: A Case Series

**DOI:** 10.3390/jcm11195495

**Published:** 2022-09-20

**Authors:** Olina Rios, Barbara Lerhe, Emmanuel Chamorey, Charles Savoldelli

**Affiliations:** 1Department of Oral and Maxillo-Facial Surgery, Head and Neck Institute, 06100 Nice, France; 2Department of Statistics, Antoine Lacassagne Center, 06100 Nice, France

**Keywords:** segmented Le Fort I osteotomy, orthognathic surgery, patient-specific implants, virtual surgical planning

## Abstract

Background: When maxillary transversal expansion is needed, two protocols of treatment can be used: a maxillary orthodontic expansion followed by a classical bimaxillary osteotomy or a bimaxillary osteotomy with maxillary segmentation. The aim of this study was to assess the accuracy of segmented Le Fort I osteotomy using computer-aided orthognathic surgery and patient-specific titanium plates in patients who underwent a bimaxillary osteotomy for occlusal trouble with maxillary transversal insufficiencies. Methods: A virtual simulation of a Le Fort I osteotomy with maxillary segmentation, a sagittal split ramus osteotomy, and genioplasty (if needed) was conducted on a preoperative three-dimensional (3D) model of each patient’s skull using ProPlan CMF 3.0 software (Materialise, Leuven, Belgium). Computer-assisted osteotomy saw-and-drill guides and patient-specific implants (PSIs, titanium plates) were produced and used during the surgery. We chose to focus on the maxillary repositioning accuracy by comparing the preoperative virtual surgical planning and the postoperative 3D outcome skulls using surface superimpositions and 13 standard dental and bone landmarks. Errors between these preoperative and postoperative landmarks were calculated and compared to discover if segmental maxillary repositioning using PSIs was accurate enough to be safely used to treat transversal insufficiencies. Results: A total of 22 consecutive patients—15 females and 7 males, with a mean age of 27.4 years—who underwent bimaxillary computer-assisted orthognathic surgery with maxillary segmentation were enrolled in the study. All patients presented with occlusion trouble, 13 with Class III malocclusions (59%) and 9 (41%) with Class II malocclusions. A quantitative analysis revealed that, overall, the mean absolute discrepancies for the x-axis (transversal dimension), y-axis (anterior–posterior dimensions), and z-axis (vertical dimension) were 0.59 mm, 0.74 mm, and 0.56 mm, respectively. The total error rate of maxillary repositioning was 0.62 mm between the postoperative cone-beam computed tomography (CBCT) and the preoperatively planned 3D skull. According to the literature, precision in maxilla repositioning is defined by an error rate (clinically relevant) at each landmark of <2 mm and a total error of <2 mm for each patient. Conclusions: A high degree of accuracy between the virtual plan and the postoperative result was observed.

## 1. Introduction

Traditionally, transversal insufficiencies are corrected by a phase of orthodontic expansions followed by a classic bimaxillary osteotomy. A segmented Le Fort I osteotomy allows for the correction of moderate transversal insufficiencies, in addition to the possibility of correcting the vertical and sagittal dimensions in a single surgery. They are useful for the correction of maxillo-mandibular asymmetries or to compensate for orthodontic insufficiencies. The outcome of this surgical technique, initially based on two-dimensional (2D) cephalometry and dental cast model analyses, is contested [1,2,3]. Proffit et al. [4,5] asserted transversal widening of the maxilla to be one of the least stable surgical corrections.

The advent of computer-assisted surgical planning and computer-aided design/computer-aided manufacturing (CAD-CAM) techniques for patient-specific implant (PSI) fabrication has enabled new methods for maxillary repositioning and evaluating planning accuracy, which are particularly complex with segmented Le Fort I movements [6,7,8,9,10,11]. PSIs can be complex three-dimensional structures such as orbital walls and parts of the maxilla [12] or saw-and-drill guides and plates [13]. They are mostly made of titanium but can be made from other materials such as PMMA, composites, or most recently, carbon-fiber-reinforced PEEK [14], which has been successfully used in the reconstruction of the skull and the facial skeleton. In this study, we only used patient-specific saw-and-drill guides and titanium plates.

Numerous authors have reported the accuracy of 3D virtually planned Le Fort I osteotomies using patient-specific implants by comparing the preoperative 3D virtually planned maxillary position with the direct postoperative maxillary position but often without differentiating between a segmented and a nonsegmented Le Fort I osteotomy. Four previous studies—da Costa Senior et al. [3], Kwon et al. [8], Stokbro et al. [15], and Ying et al. [16]—focused on the accuracy of maxillary repositioning with segmented Le Fort I osteotomies, but they used surgical splints [8,15,16] or classical plates [3]. The conclusion of Stokbro et al. [15] was that not all of the planned expansion was obtained during segmental bimaxillary surgery using surgical splints. To the best of our knowledge, no previous study has assessed the accuracy of maxillary repositioning with a bi- or tripartite disjunction using PSIs.

According to Juho Suojanen, PSIs have been demonstrated to be an ideal solution for deformity surgery and the reconstruction of complex post-traumatic cases. Their use in classic Le Fort I osteotomies is spreading, and they represent an interesting tool for complex maxillary movements in orthognathic surgery, including multisegmental osteotomies [13]. Patient-specific cutting guides with predictive plate holes and three-dimensional printed plates have allowed for accurate nonsegmented maxillary repositioning based only on the bone-borne patient-specific guide and plate [13,17,18,19,20,21]. This eliminates the need for intermediate splints. It also appears to decrease the surgery duration.

Recently, Greenberg et al. [20] studied the accuracy of patient-specific surgical guides and titanium plates in the maxillary repositioning of 10 patients. The median absolute discrepancies between the postoperative position and the planned position on the x-axis, y-axis, and z-axis were 0.638, 0.798, and 0.481 mm, respectively. In this study, the case with the most discrepancies when comparing the postoperative result with the planned surgery was the only one who underwent a tripartite disjunction (a three-piece maxillary segmentation). Two hypotheses were formulated to explain those discrepancies: (1) the unpredictable soft tissue reaction to segmental and extension movements, particularly the relative inextensibility of the fibro-palatine mucosa, which acts in opposition to transversal widening, and (2) the lack of stability of segmental maxillary osteotomies.

Therefore, the aim of this study was to assess the accuracy of segmented Le Fort I osteotomies using computer-aided orthognathic surgery and patient-specific titanium plates in patients who underwent a bimaxillary osteotomy with a maxilla-first protocol for occlusal trouble.

## 2. Materials and Methods

### 2.1. Study Design

This was a retrospective case series of 22 patients who underwent bimaxillary orthognathic surgery (Le Fort I osteotomy with a bi- or tripartite disjunction and a mandibular bilateral sagittal split osteotomy (BSSO)) with one-piece bone-borne maxillary patient-specific surgical cutting guides and one-piece maxillary plates at the Head and Neck Institute, Nice, France, between September 2017 and July 2021.

### 2.2. Study Sample

This study was conducted in accordance with the World Medical Association Declaration of Helsinki on medical research and with the approval of the Face and Neck Institute Ethics Committee. Conforming with data protection regulations, every patient was informed of this study and the use of their data; none of them refused to participate. All patient data were anonymized before the analysis.

The inclusion criteria were patients undergoing a two-piece (median or asymmetric) or three-piece Le Fort I osteotomy and access to preoperative and postoperative (one week after surgery) cone-beam computed tomography (CBCT) images. The exclusion criteria were composed of cleft and syndromic patients and a history of maxillo-facial trauma (Figure 1). The population characteristics are listed in Table 1.

### 2.3. Set-Up

During the presurgical workup, photographs, dental impressions, and CBCT studies of the patients were obtained. To acquire a preoperative 3D skull, we used numerical tooth prints obtained from Carestream 9600 (Carestream Dental, Atlanta, GA, USA) and preoperative standardized CBCT [22,23,24,25,26,27] one month before surgery (field of view: 22 × 16 cm; scan time: 40 s; voxel size: 0.4 mm) (I-CAT 3D imaging System, Imaging Sciences International, Inc., Hatfield, PA, USA). The CBCT data were exported to digital imaging and communications in medicine format (DICOM).

### 2.4. Virtual Surgical Planning

All data recorded during the set-up were sent to Materialise (Malakoff, France). All cases included in this study were virtually planned using ProPlan CMF 3.0 software (Materialise, Leuven, Belgium) by the same clinical engineer. The virtual planning was then obtained from a web meeting (Figure 2).

### 2.5. Surgical Technique

All bimaxillary osteotomies were performed using the same surgical technique with a maxilla-first protocol. The incision was made from the left maxillary first molar to the right maxillary first molar. After periosteum dissection and maxilla exposure, the patient-specific maxillary cutting guide was placed and tied using two 2 mm screws, as shown in Figure 2.

Using predrilling cylinders, predictive holes were made. Each hole corresponded with the screw location on the final maxillary plate. Using a piezoelectric unit, the Le Fort I osteotomy was marked. Then, the maxillary guide was removed, and the osteotomy was finalized with a reciprocating saw and osteotomes, allowing the maxilla to be downfractured. The bi- or tripartite disjunction was achieved using a piezoelectric unit (Figure 3). In the case of a three-piece Le Fort I osteotomy, the osteotomy was performed distal to the lateral incisor or to the canine, according to the planning and patient-specific needs.

Finally, the maxillary plate was fixed to the downfractured maxilla with 5 mm screws. As a result, the maxilla was passively repositioned without the need for a dental splint or reference markings. The rest of the procedure followed the same principle with a patient-specific mandibular guide and two patient-specific plates (three if a genioplasty was needed). At the conclusion of the procedure, the final occlusion was visually evaluated. All occlusions were visually similar to the planned occlusions.

### 2.6. Study Variables

The primary outcome of interest was the postoperative position of the maxilla, which was assessed by CBCT one week after surgery. This postoperative position was compared with the virtually planned postoperative position of the maxilla using a surface superimposition (in all cases, a composite 3D model of the cranium was created by superimposing CBCT images) and 13 standard dental and bone landmarks evaluated in three dimensions. This allowed for the visualization and measurement of the discrepancies between the planned and postoperative maxillary positions (Figure 4).

Cephalometric landmarks were recorded on the planned maxilla and postoperative maxilla (Table 2). Those points were placed on a 3D Cartesian coordinate system representing the three anatomic axes: the x-axis (right–left direction), the y-axis (anterior–posterior direction), and the z-axis (superior–inferior direction).

The discrepancies in each planned surgical movement at each cephalometric landmark in all three planes were calculated by subtracting the postoperative maxillary coordinates from the planned maxillary coordinates. The absolute values of these discrepancies were tabulated. We also chose to study two types of distances: (1) the distances between the dental landmarks and the Frankfort plane, to evaluate the difficulties imposed by bony interferences during impaction, and (2) the distances between the dental landmarks, to emphasize the spacing created by the maxillary disjunction and evaluate the difficulties imposed by the relative inextensibility of the fibro-palatine mucosa (Figure 5). All the landmarks were identified twice by the same clinical engineer.

### 2.7. Statistical Analysis

The statistical analysis was performed by an independent statistician who was not an investigator in this study.

## 3. Results

Twenty-two patients consisting of ten two-piece Le Fort I osteotomy cases, nine three-piece cases, and three asymmetric two-piece cases were included in the study between September 2017 and July 2021, following the inclusion/exclusion criteria. There were 15 females and 7 males with a mean age of 27.4 years. The total error rate of the maxillary repositioning at each landmark across the three axes can be found in Table 3.

The median planned surgical movement was a 1.19 mm expansion on the x-axis (right–left direction), a 2.71 mm expansion on the y-axis (anterior–posterior direction), and a 1.50 mm impaction on the z-axis (superior–inferior direction).

According to the literature, precision in maxilla repositioning is defined by the error rate (clinically relevant) at each landmark of <2 mm and the total error of <2 mm for each patient [10,24,28]. Overall, the mean absolute discrepancies for the x-axis (transversal dimension), y-axis (anterior–posterior dimensions), and z-axis (vertical dimension) were 0.59 mm, 0.74 mm, and 0.56 mm, respectively. The total error rate of maxillary repositioning was 0.62 mm. The surgical discrepancies were similar for the two- and three-piece segmental maxillary osteotomies. This demonstrates a high precision in the positioning of the maxilla, as it was calculated using virtual planning and PSIs. No statistically significant differences were found in the average absolute discrepancies between the x-axis and the y-axis (*p* = 0.37), between the x-axis and the z-axis (*p* = 0.48), or between the y-axis and the z-axis (*p* = 0.11) after Fisher’s exact tests. This suggests that the patient-specific guides and plates were equally accurate in all three dimensions. Boxplots were created to illustrate the discrepancies at each landmark (Figure 6).

## 4. Discussion

The aim of this study was to assess the accuracy of patient-specific guides and plates in segmented maxillary repositioning during bimaxillary orthognathic surgery. Calculating the error rate in positioning the maxilla at each anatomical landmark allowed us to determine that (1) the patient-specific cutting guides and plates are positioned on the maxilla according to the surgical plan with a high level of accuracy and (2) the segmented postoperative maxilla is repositioned conformally to planification.

In the study by Heufelder et al. [19], an overall 3D discrepancy of 0.85 mm was measured. Comparably, in the study by Greenberg et al. [20], an overall 3D discrepancy of 1.129 mm was obtained. Those studies compared all bimaxillary osteotomies without a subanalysis of segmental osteotomies. In the study by Kwon et al. [8], which focused on segmental Le Fort I osteotomies with the use of surgical splints, the absolute mean linear difference was 0.96 mm transversely, 1.23 mm vertically, and 1.16 mm anteriorly–posteriorly. In our study, our results were 0.59 mm transversely, 0.56 mm vertically, and 0.74 mm anteriorly–posteriorly. The results of our study were similar to other studies assessing the precision of maxillary repositioning using PSIs. However, to the best of our knowledge, our study is the first to assess the accuracy of segmented maxillary repositioning with PSIs. It is also the only study that assesses the accuracy of asymmetric disjunctions.

Segmented Le Fort I osteotomies seem to be more frequently required in patients with Class III malocclusions, often resulting from transverse maxillary deficiency or asymmetry. Even if Class II malocclusions are the most frequent dentofacial deformities worldwide [29,30,31], patients with Class III malocclusions seem to seek maxillo-facial surgery more frequently because of higher aesthetic or functional demands [32,33]. Furthermore, they are more often treated with a bimaxillary osteotomy, indicating the higher prevalence of skeletal discrepancies affecting both jaws in Class III subjects [34].

Our results presented the following question: how can we explain the 0.62 mm discrepancy between what was planned and what was observable on the postoperative CBCT?

Our first hypothesis was a lack of accuracy in the landmark measurements. Manual landmark identification using 3D cephalometry is the most frequently used method to quantify maxillary surgical movements in the literature [6,13,17,18,19,20,21]. The major limitation of this method is the imprecision caused by the manual identification of the landmarks [35,36,37,38], which may be affected by human error in the range of 0.3–2.8 mm [24]. Moreover, this technique is relatively time-consuming because numerous landmarks must be identified multiple times.

To overcome landmark identification errors, the following two methods are currently being studied: fully automatic landmark recognition [37,38,39] and the elimination of landmark-based measurements [35,36]. Baan et al. [40] developed the OrthoGnathicAnalyser (OGA), a semi-automatic approach to assess the accuracy of bimaxillary osteotomies. The OGA uses voxel-based registration (VBR) technology, relying on volumetric data rather than the surfaces of 3D objects or landmarks. Recently, Xi et al. compared the accuracy of landmark-based versus voxel-based 3-dimensional (3D) analysis to quantify maxillo-mandibular movements after bimaxillary osteotomy. High intraclass correlation coefficients were found for both methods. However, the voxel-based analysis yielded a higher correlation concerning the maxilla and the distal mandible [35]. To date, the OGA has not yet been validated for the assessment of segmental Le Fort I osteotomies, although the study by da Costa Senior et al. [3] appears to be encouraging.

Some authors assert that voxel-based registration, first introduced by Cevidanes et al. in 2005 [28,41,42], is the most accurate method to assess jaw movements. However, in a study by Al-Mukhtar et al. [43], no statistical significance was found between the surface and voxel-based registration methods.

One strength of this study was the precocity of a CBCT control (one week after surgery), which illustrated an almost immediate postoperative result (and, by extension, the operator precision). This was also a weakness because it did not demonstrate the real maxilla position after bone consolidation (in many studies, this CBCT control is undertaken after two months), which could illustrate the material’s efficiency to resist all bone and soft tissue constraints.

The principle of PSIs is that the plates themselves impose a maxillary movement that was preoperatively chosen during surgical planning. The issue remains as to whether they are strong enough to prevent their own movement under the forces imposed by the bones and facial muscles during consolidation. It is believed that the shape of a one-piece maxillary plate acts as a cage around the maxilla and thus reinforces the maxilla stability, making the use of distractors or contention methods unnecessary.

To assess the plate movements, we focused on the plate superimposition of our less precise case by subtracting its postoperative position from its planned position. The plate seemed to be subject to a greater number of constraints at its extremities without a significant deformation to its structure (Figure 7).

Recently, Holte et al. [44] assessed segmental bimaxillary surgery stability with long-term cone-beam computed tomography two years after surgery. All of their results were within a clinically acceptable stability of 2 mm. This is an indirect indicator of good plate stability; however, they used classical plates. Further investigations are needed to evaluate the discrepancies in postoperative patient-specific plate positions after several years.

Third, surgical experience teaches us that difficulties in repositioning the maxilla are the greatest in the transversal plane (due to the inextensibility of the fibro-palatine mucosa) and the vertical plane (due to posterior bone interferences during impaction). This led us to the conclusion that our error rate would be maximal in those directions, thus motivating an evaluation by a comparison with a Frankfort plane (vertical interferences) and between the dental landmarks (transversal expansion). We now understand that we were not less precise in those planes, and PSIs appear to be an effective method for complex maxillary movements, as in segmented Le Fort I osteotomies. However, the sources of the surgical discrepancies require further analysis to improve their accuracy.

## 5. Conclusions

In this study, patient-specific implants seem to be an accurate material for segmental Le Fort I osteotomies. Further studies are needed to assess the long-term stability of these types of plates and surgery.

## Figures and Tables

**Figure 1 jcm-11-05495-f001:**
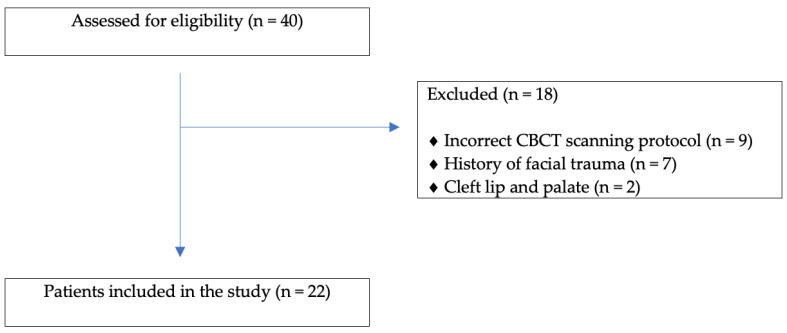
Flowchart.

**Figure 2 jcm-11-05495-f002:**
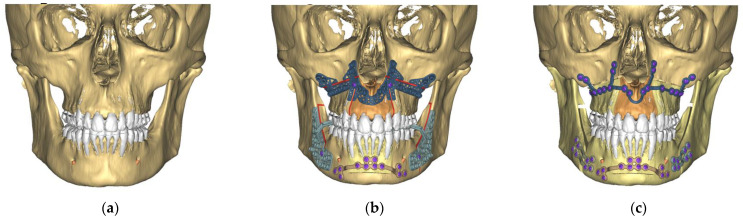
Model of a patient’s head (**a**), virtual planning of the patient-specific guides (**b**), and virtual planning of the patient-specific plates (**c**).

**Figure 3 jcm-11-05495-f003:**
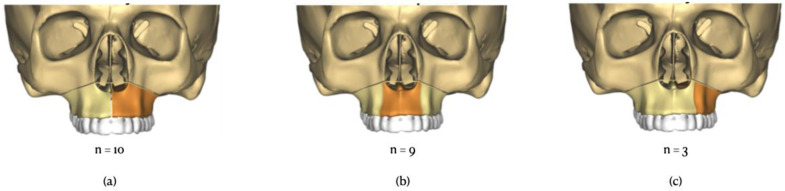
Types of maxillary segmentation. (**a**) Two-piece Le Fort I osteotomy with median disjunction, (**b**) three-piece Le Fort I osteotomy, (**c**) two-piece Le Fort I osteotomy with asymmetric disjunction (2 left, 1 right).

**Figure 4 jcm-11-05495-f004:**
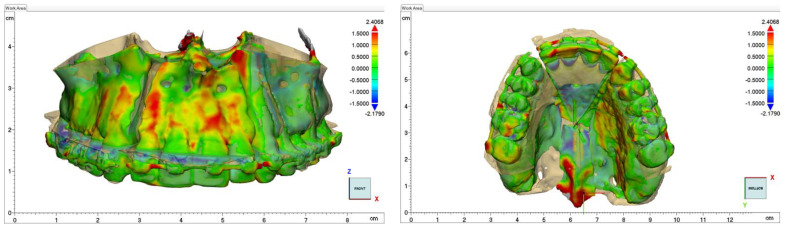
Focus on maxillary discrepancies with color-coded isovalor modifications obtained by superimposing postoperative CBCT with virtually planned postoperative maxilla. The scale is in millimeters.

**Figure 5 jcm-11-05495-f005:**
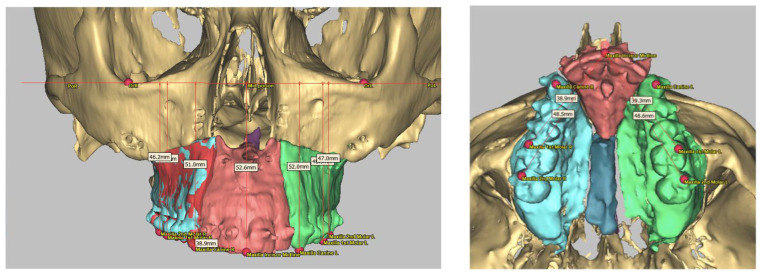
Measures with Frankfort plane and interdental spaces.

**Figure 6 jcm-11-05495-f006:**
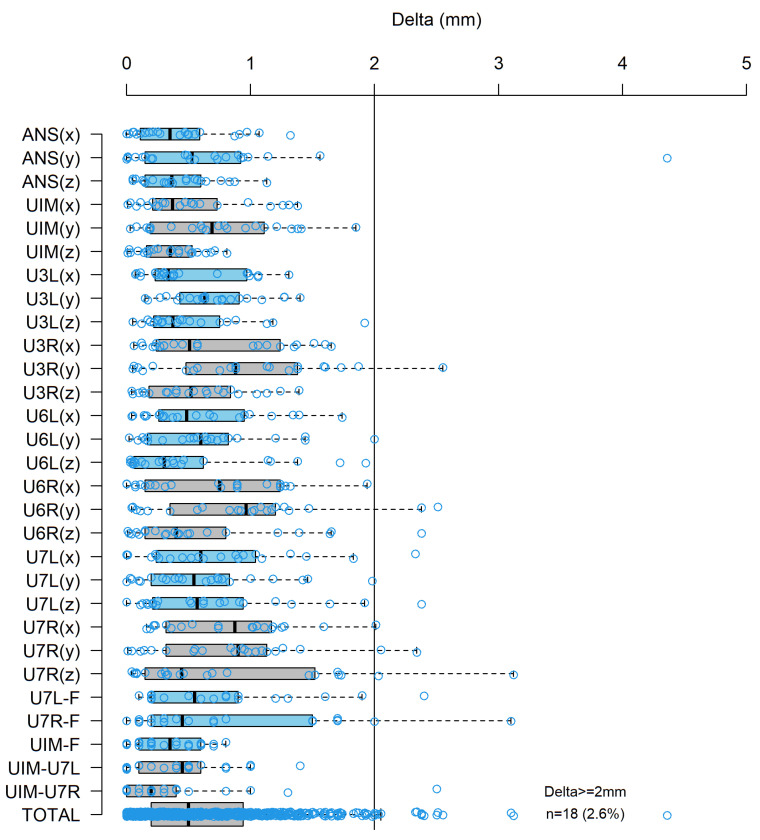
Boxplots presenting the absolute discrepancy between the postoperative position and the virtually planned postoperative position of the maxilla at each reference point.

**Figure 7 jcm-11-05495-f007:**
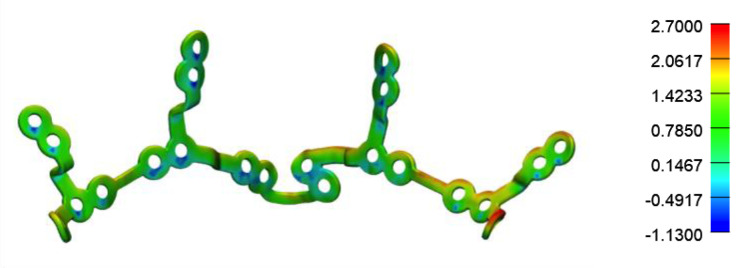
Plate superimposition.

**Table 1 jcm-11-05495-t001:** Study sample characteristics.

Variables	
Mean age (years)	27.4
Range	15–52
Sex	15 F 7 M
Disjunction type	
Median	10
Tripartite	9
Asymmetric	2 left, 1 right

**Table 2 jcm-11-05495-t002:** Landmarks.

Landmark	Correspondence
ANS	Anterior nasal spine (x, y, z)
UIM	Most mesial point of the tip of the crown in between the upper central incisors (x, y, z)
U3L	The most inferior point of the tip of the crown of the upper left canine (x, y, z)
U3R	The most inferior point of the tip of the crown of the upper right canine (x, y, z)
U6L	The most inferior point of the mesial buccal cusp of the crown of the first upper left molar (x, y, z)
U6R	The most inferior point of the mesial buccal cusp of the crown of the first upper right molar (x, y, z)
U7L	The most inferior point of the mesial buccal cusp of the crown of the second upper left molar (x, y, z)
U7R	The most inferior point of the mesial buccal cusp of the crown of the second upper right molar (x, y, z)
U7L–F	Maxilla second molar left with Frankfort plane
U7R–F	Maxilla second molar right with Frankfort plane
UIM–F	Maxilla incisor midline with Frankfort plane
UIM–U7L	Maxilla incisor midline to second left molar
UIM–U7R	Maxilla incisor midline to second right molar

**Table 3 jcm-11-05495-t003:** Total error rate at each landmark, in millimeters. The second column represents the mean linear discrepancies between the postoperative planned position of each landmark and the obtained postoperative position of each landmark, assessed by CBCT one week after surgery.

Landmarks	Mean (SD)	Median (Min–Max)
ANS(x)	0.43 (0.38)	0.35 [0–1.3]
ANS(y)	0.72 (0.92)	0.53 [0–4.4]
ANS(z)	0.43 (0.3)	0.36 [0.05–1.1]
UIM(x)	0.52 (0.44)	0.37 [0.01–1.4]
UIM(y)	0.74 (0.51)	0.69 [0.03–1.8]
UIM(z)	0.35 (0.24)	0.36 [0.01–0.81]
U3L(x)	0.5 (0.39)	0.34 [0.07–1.3]
U3L(y)	0.7 (0.35)	0.63 [0.15–1.4]
U3L(z)	0.52 (0.45)	0.38 [0.05–1.9]
U3R(x)	0.72 (0.55)	0.51 [0.06–1.6]
U3R(y)	0.96 (0.68)	0.88 [0.05–2.6]
U3R(z)	0.57 (0.4)	0.52 [0.04–1.4]
U6L(x)	0.62 (0.48)	0.48 [0.04–1.7]
U6L(y)	0.65 (0.51)	0.6 [0.02–2]
U6L(z)	0.5 (0.58)	0.3 [0.03–1.9]
U6R(x)	0.68 (0.56)	0.75 [0–1.9]
U6R(y)	0.92 (0.67)	0.97 [0.04–2.5]
U6R(z)	0.61 (0.65)	0.4 [0.01–2.4]
U7L(x)	0.73 (0.61)	0.6 [0–2.3]
U7L(y)	0.63 (0.53)	0.55 [0,1,2]
U7L(z)	0.71 (0.64)	0.57 [0–2.4]
U7R(x)	0.81 (0.51)	0.88 [0.16–2]
U7R(y)	0.85 (0.61)	0.9 [0.01–2.3]
U7R(z)	0.82 (0.84)	0.44 [0.04–3.1]
U7L–F	0.72 (0.62)	0.55 [0.1–2.4]
U7R–F	0.82 (0.83)	0.45 [0–3.1]
UIM–F	0.35 (0.25)	0.35 [0–0.8]
UIM–U7L	0.44 (0.39)	0.45 [0–1.4]
UIM–U7R	0.39 (0.59)	0.2 [0–2.5]
TOTAL	0.62 (0.57)	0.48 [0–4.4]

## Data Availability

Not applicable.

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
