# Peer review of "Accuracy of Segmented Le Fort I Osteotomy with Virtual Planning in Orthognathic Surgery Using Patient-Specific Implants: A Case Series"

_jcm, 2022, doi:10.3390/jcm11195495_

Round 1
Reviewer 1 Report
Dear authors,
Congratulations on your work!
The paper Accuracy of segmented Le Fort I osteotomy by virtual planning 2 in orthognathic surgery using patient-specific implants assessed the accuracy of segmented Le Fort I osteotomy using computer- 11 aided orthognathic surgery and patient-specific titanium plates in patients who under- 12 went bimaxillary osteotomy for occlusal trouble with maxillary transversal insufficiencies.
Keywords are appropiate.
Abstract is well written.
Introduction provides up to date information and is well written.
The virtual surgery planning is perfectly described.
Figure 2. Model of the patient head (a), virtual planning of the patient-specific guides (b), virtual 120 planning of the patient-specific plates (c). and Figure 3. Types of maxillary segmentation are very clear.
Figure 5 : Mesures with Frankfort Plan and interdental spaces is good described and adds enefit to the value of the study.
Results are clear wtitten adn discussion fosuses on relavant aspects, such as how can we explain the discrepancy between what was planned and what was observable on the postoperative CBCT.
Authors discuss how to overcome landmark identification errors.
Aslo strenghts and limitations are debated.
Conclusion is clear and sustainable.
Author Response
Dear Reviewer,
I am writing to confirm that I well received your comments.
First of all, I thank you very much for reading my work and reviewing it. I appreciate all your kinds comments about each section of this article. I am glad you find my article clear, interesting and appropriately done.
Your words really encourage me for my future researches.
I humbly remain at your disposal if any modification is needed.
Thank you again for according me your time and consideration, Best regards,
Olina RIOS

Reviewer 2 Report
Dear Authors,
first of all, thank you for giving me the opportunity to review your manuscript.
Below you will find some indications to improve its drafting.
Title:
indicate the type of study you performed
Abstract:
review the drafting of the summary, also report any software and statistical analyzes used as well as the significance values.
Introduction:
line 54: report the reference here.
Aim: you talk about bimaxillary osteotomy; specify also here that you used maxilla first technique. Do you believe that the different protocols, i.e. mandible first, can affect the results in terms of accuracy?
Materials and methods:
Figure 1 must be revised in accordance with the CONSORT-type guidelines.
Table 1: correctly specify the variables (Mean Age (years) 27.4 and Range 15-52)
Results:
table 3: why did you report the median? Bringing back the data correctly collated makes it easier to read.
Discussion:
this part needs to be revised and improved because, at present, it is difficult to understand.
A suggestion: evaluate a comparison by means of 3D superimposition and calculation of the mean square error of surfaces.
Thanks for your work and good luck!
Author Response
Dear Reviewer,
I am writing to confirm that I well received all your comments.
First of all, I thank you very much for reading my work and reviewing it. I appreciate your remarks to improve it and I did my best to modify the article according to them. Please see the attachment. Best regards, Olina RIOS
Reviewer 3 Report
Paper needs revision for grammar and style, many typos
Abstract, add the mean age for patients, male/female, the type of malocclusion that they had, and add the name of the software that you used
Introduction, patient-specific implants (PSIs) is a very vague term, find a better terminology, explain what you exactly mean by PSI,
2nd page, 2nd paragraph, there are many sentences without citation
2nd page, 3rd paragraph, revise ' Hypotheses can be proposed to explain those discrepancies: (1) the unpredictable soft tissue reaction to segmental and extension movements, particularly the relative inextensibility of the fibro-palatin mucosa; and (2) the lack of stability of segmental maxillary osteotomies.'
Results, divide the difference in Anterior-posterior, Vertical, and transverse dimensions, which was the highest difference, was there any difference between single and double jaw surgery in terms of accuracy?
Add a citation for 'According to the literature, precision in maxilla repositioning is defined by an error rate (clinically relevant) at each landmark < 2 mm and a total error < 2 mm for each patient. '
Figure 3, add more information on the type of maxillary sectioning, similarly for table 3
Discussion, expand, and add that this type of surgery is more common in Class III cases (with more transverse deficiency or asymmetry that needs segmentation or expansion of maxilla) and they form majority of orthognathic surgery cases(read, cite, J Plast Reconstr Aesthet Surg. 2016 Jun;69(6):796-801. ;Int J Environ Res Public Health. 2019 May 28;16(11):1881.;Ann Maxillofac Surg. 2017 Jan-Jun;7(1):73-77. )
Author Response
Dear Reviewer,
I am writing to confirm that I well received your comments.
First of all, I thank you very much for reading my work and reviewing it. I appreciate your remarks to improve it and I did my best to modify the article according to them.
Please see the attachment.
Thank you for according me your time and consideration,
Best regards,
Olina RIOS

Round 2
Reviewer 2 Report
Dear Authors,
thanks for your full revision; no more modifications are required.
Good luck!
Author Response
Dear Reviewer,
Thank you very much for reading and reviewing my work again. I am glad to notice my corrections are satisfying.
Thank you again for according me your time and consideration.
Best regards,
Olina RIOS

Reviewer 3 Report
Thank you for the revision, paper still needs revision for English and some terminologies that you used
For instance, instead of ' type III malocclusions 29 (59%) and 9 (41%) with type II malocclusion' use 'Class III malocclusions 29 (59%) and 9 (41%) with Class II malocclusions'
or ', patients with type III malocclusions seems to consulted 332 more often for maxillo-facial surgery because of' change to ', patients with Class III malocclusions seem to seek more frequently maxillo-facial surgery because of'
line 266-276, revise please
line 269, You say 'Overall, the mean absolute discrepancies for the x-axis, y-axis and z-axis were 0.59 mm, 0.74 mm and 0.56 mm, respectively', Do you mean 'Overall, the mean differences for the x-axis (vertical dimension), y-axis (anterior-posterior dimensions) and z-axis (transverse dimension) were 0.59 mm, 0.74 mm and 0.56 mm, respectively' , you also need to add this information to the abstract
Author Response
Dear Reviewer,
Thank you very much for reading and reviewing my work again.
Please see the attachment.
Best regards,
Olina RIOS
